# Development of HER2-Specific Aptamer-Drug Conjugate for Breast Cancer Therapy

**DOI:** 10.3390/ijms21249764

**Published:** 2020-12-21

**Authors:** Hwa Yeon Jeong, Hyeri Kim, Myunghwa Lee, Jinju Hong, Joo Han Lee, Jeonghyeon Kim, Moon Jung Choi, Yong Serk Park, Sung-Chun Kim

**Affiliations:** 1Biois Co., Ltd., Seoul 08390, Korea; fhwayeon@gmail.com (H.Y.J.); khl891230@naver.com (H.K.); myunghwa_s2@naver.com (M.L.); hjz8807@naver.com (J.H.); blucian00@naver.com (J.H.L.); jeonghyeonkim9@gmail.com (J.K.); 2Department of Biomedical Laboratory Science, Yonsei University, Wonju 26493, Korea; tt1651@naver.com

**Keywords:** HER2-positive cancer, RNA aptamer, mertansine, aptamer delivery

## Abstract

In this study, HER2 RNA aptamers were conjugated to mertansine (DM1) and the anti-cancer effectiveness of the conjugate was evaluated in HER2-overexpressing breast cancer models. The conjugate of HER2 aptamer and anticancer drug DM1 (aptamer-drug conjugate, ApDC) was prepared and analyzed using HPLC and mass spectrometry. The cell-binding affinity and cytotoxicity of the conjugate were determined using confocal microscopy and WST-1 assay. The in vivo anti-tumoral efficacy of ApDC was also evaluated in mice carrying BT-474 breast tumors overexpressing HER2. The synthesized HER2-specific RNA aptamers were able to specifically and efficiently bind to HER-positive BT-474 breast cancer cells, but not to HER2-negative MDA-MB-231 breast cancer cells. Also, the HER2-specific ApDC showed strong toxicity to the target cells, BT-474, but not to MDA-MB-231 cells. According to the in vivo analyses drawn from the mouse xenografts of BT-747 tumor, the ApDC was able to more effectively inhibit the tumor growth. Compared to the control group, the mice treated with the ApDC showed a significant reduction of tumor growth. Besides, any significant body weight losses or hepatic toxicities were monitored in the ApDC-treated mice. This research suggests the HER2 aptamer-DM1 conjugate as a target-specific anti-cancer modality and provides experimental evidence supporting its enhanced effectiveness for HER2-overexpressing target tumors. This type of aptamer-conjugated anticancer drug would be utilized as a platform structure for the development of versatile targeted high-performance anticancer drugs by adopting the easy deformability and high affinity of aptamers.

## 1. Introduction

Targeted therapy is an essential procedure for precision and personalized therapy to improve patient care [1]. In order to bring target-specificity, varied target ligands are coupled to therapeutic drugs by various linking strategies such as direct coupling, via biocompatible spacers, cleavable linkers, or other payload chemistries [2,3,4]. In all these cases, the efficiency and safety of the conjugate are primarily determined by the nature of payload and targeting ligand. Compared to untargeted therapies, for example, chemotherapy or radiotherapy of cancer, targeted drug delivery intends to deliver toxic therapeutic cargos specifically into diseased cells and avoids off-target uptake by healthy normal cells [5,6,7]. To find adoptable targeting ligands, a variety of small organic molecules, proteins, and nucleic acid scaffolds have been extensively searched based on the biomarkers selected [8,9].

Among ligands for targeted therapy, antibodies are well-established for specific recognition and/or biological regulation, and, therefore, antibody−drug conjugates (ADCs) have attracted tremendous attention for targeted cancer therapy [10]. Besides, certain structures of aptamers and peptides have been also recognized as a useful targeting moiety for targeted therapy [11]. Recently, exciting advances in oncology have been realized from an old idea about clinical applications of ADCs which was first tested 30 years ago [12]. Brentuximab vedotin (anti-CD30–monomethyl auristatin E, MMAE) and ado-trastuzumab emtansine (anti-HER2–DM1), prepared through improved antibody–drug linker chemistry, were recently approved by the Food and Drug Administration (FDA) for cancer treatment [13,14]. The conjugate of a prostate-specific membrane antigen (PSMA) antibody (SCL44A4) and highly toxic monomethyl auristatin E (MMAE) has been also testing as a target-specific ADC [15,16].

Separately, various structures of aptamers have been developed as targeting ligands. Aptamers which are single-stranded oligonucleotides with specific recognition capabilities to target molecules are primarily screened through the Systematic Evolution of Ligands by Exponential Enrichment (SELEX) [17,18,19,20] in combination with purification and analysis techniques such as capillary electrophoresis, microfluidics, and fluorescence-activated cell sorting [21]. The molecular targets of aptamers range from small molecules and proteins to intact cells. A wide array of aptamers with high binding affinities (Kds of pM-nM) has been developed for biomarkers of high therapeutic interest. Recent advances in biochemistry and biophysics have facilitated the resolution of crystal structures of aptamer−target complexes which brings elucidation of aptamer−target interactions in detail. This facilitates the development of aptamers and aptamer-drug conjugates (ApDCs) through rational drug design and screening, as well as the discovery of new druggable sites on biomarkers [22].

In addition to target binding, aptamers can also act as therapeutics to regulate the biological functions of target molecules. For example, pegaptanib (marketed as Macugen®) [23], an anti-VEGF aptamer that recognizes most human VEGF-A isoforms, has been approved by the FDA for the treatment of age-related macular degeneration (AMD). Another example, AS1411, is a nucleolin-targeting aptamer currently under Phase II clinical investigation for the treatment of acute myeloid leukemia (AML) [24]. The several advantages over antibody such as low immunogenicity, rapid tissue penetration, cost-effectiveness in synthesis and modification, and long shelf life accelerate their clinical applications as therapeutics and targeting ligands [25].

ErbB family is one of the most popular receptors for targeted therapies. In particular, EGFR (ErbB1 and HER1) are associated with a variety of solid tumor malignancies and the overexpression of ErbB2 (HER2) is also found in 20–30% of breast cancers [26]. To date, synthetic tyrosine kinase inhibitors (e.g., Erlotinib and Gefitinib) and monoclonal antibodies (mAbs; e.g., Cetuximab and Trastuzumab) have been developed to inhibit certain signaling pathways in cancer cells or recruit the immune system to cancer cells [27]. Several anti-ErbB2 aptamers were already developed as an alternative therapeutic modality [28]. The anti-ErbB2 RNA aptamer was used in this study for ApDC development [29].

In this study, we developed a HER2-specific aptamer-drug conjugate for targeted cancer therapy. The HER2-specific aptamer was conjugated with anti-cancer mertansine (DM1) by a cleavable disulfide bond at the 3′ end of aptamer [30,31,32]. A polyethylene glycol molecule (M.W. 20 kDa) was also conjugated to the 5′ end of aptamer for prolonged circulation in vivo. The HER2-specific ApDC was tested in HER2-positive cancer cells to verify its target-specific binding capability and cytotoxicity. Also, the ApDC was systemically administered into mice carrying tumor xenografts to evaluate its targeted anti-cancer therapeutic efficacy.

## 2. Results and Discussion

### 2.1. Characterization of HER2 RNA Aptamer

It is known that RNA molecules are generally unstable and easily degraded by RNases in various environments. However, the stability of RNA can be secured by various conventional modification methods, and the RNA aptamer used in the present study (Figure 1A) was also modified at the pyrimidine by 2′-fluorinated pyrimidine, thereby achieving much higher stability than the unmodified RNA aptamer [33]. The gel retardation analysis of aptamer molecules (Figure 1B) shows the relative stabilities of unmodified RNA aptamer and 2′-pyrimidine-modified RNA aptamer in 10% FBS-containing medium. The results of gel retardation analysis of the unmodified RNA aptamer revealed that the unmodified molecules degraded more than 50% in 1 h. On the other hand, 2′-pyrimidine-modified RNA aptamers were able to maintain their structure over 50% even 48 h later.

The 2′-pyrimidine-modified RNA aptamers utilized in this study exhibited a strong binding affinity to HER2 receptors. According to the qPCR measurement, the dissociation constant of the HER2 aptamer was 16.5 ± 4.4 nM (Figure 1C). Their HER2-specific binding was also confirmed by microscopic assessment (Figure 1D). Fluorescent HER2 aptamers effectively bind to BT-474 and A549 cancer cells, both over-expressing HER2 receptors. Meanwhile, the same molecules little bind MCF7 low-expressing HER2 and MDA-MB-231 cells deficient of HER2. These results suggest that the 2′-pyrimidine-modified HER2 RNA aptamer prepared for this study can be utilized as a ligand for HER2-specific delivery of anticancer drugs in vivo.

### 2.2. Chemistry of HER2 Aptamer-DM1 Conjugate

For the synthesis of HER2 aptamer-drug conjugate (ApDC), a 20 kDa PEG molecule was coupled to the 5′ end of HER2 RNA aptamer and a molecule of DM1 was then attached to the other side 3′ end (Figure 2A). DM1 is a widely utilized anticancer drug and has a structural advantage at the coupling to carrier molecules since it is a thiol form of mertansine. The disulfide bond between DM1 and the 3′ end of the aptamer is expected to be broken in the reducing environment of early and late endosomes. The linkage fragility of ApDC could facilitate the release of DM1 from the endocytic pathway, elevating the anticancer therapeutic efficacy of the drug [30,34,35]. Also, PEGylation of the aptamer at 5′ end could enhance the half-life of ApDC in circulation by avoiding renal clearance [36]. The prolonged circulation of the ApDC is expected to increase the availability and targetability of ApDC molecules to target cancer cells, presumably resulting in increased anticancer therapeutic efficacy [37].

According to the HPLC analysis (Figure 2B) and mass spectrometry (Figure 2C), the final product of ApDC exhibited 98% of conjugation yield and ~33,000 g/mole of molecular weight. The strict and stable procedure of ApDC synthesis provide strong advantages over biological ligand conjugates in terms of structural modification and quality control. Because of the stability of aptamer chemistry, there was little variation from batch to batch and their derivatives such as aptamer-drug conjugates were able to maintain targetability as well as therapeutic efficacy.

### 2.3. In Vitro Cytotoxicity of ApDC

The target-specific cytotoxicity of HER2 aptamer-DM1 conjugate was analyzed by the WST-1 assay (*n* = 8) on HER2-positive BT-474 and HER2-negative MDA-MB-231 cancer cells. In the target BT-474 cancer cells, free DM1 and the HER2 ApDC exhibited similar levels of IC50 value, 230.4 nM and 201.7 nM, respectively (Figure 3A). However, the MDA-MB-231 cells deficient of HER2 did not respond to the HER2 ApDC while the same cells showed cytotoxic response to the free DM 1 (Figure 3B). The ApDC hardly affect the viability of MDA-MB-231 even at the highest concentration tested (2 μM). This implies that the cytotoxicity of the ApDC is dependent upon the HER2 expressed on the surface of cancer cells.

The free form of DM1 was able to penetrate the plasma membranes regardless HER2 expression. Therefore, it can be stated that the ApDC has cell-specific cytotoxicity. Presumably, the aptamer-drug conjugates were too big and too hydrophilic to directly penetrate through the plasma membranes via a certain transport and appeared to be internalized via HER2-mediated endocytosis. The disulfide bond was easily cleaved in the in the reducing environment of endocytic vesicles, resulting in facilitated escapes of the freed DM1 from endosomes [30,34,35]. Free HER aptamers did not show any cell toxicity even to BT-474 cells at the highest concentrations. This implies the cytotoxicity of the ApDC solely comes from the drug DM1, not from the HER2 aptamer.

### 2.4. In Vivo Tumor Growth Inhibition by ApDC

In order to examine the antitumoral activity of the HER2 aptamer-DM1 conjugate, the HER2 ApDC was intravenously administered to mice carrying BT-474 tumors three times at intervals of 3 days. Tumor volumes and body weights were measured for a total of 36 days. Compared with the saline-treated group of mice, both the free DM1 and the ApDC exhibited a reduction of tumor size just after treatment (Figure 4A) and then more effectively suppressed the growth of the tumor. When the tumor-carrying mice were treated with the same amount of DM1 (60 μg/kg), the tumors of mice injected with the ApDC were statically smaller than those with free DM1 after the 19th-day post-treatment. These results tell that the chemically modified DM1 is still effective in inhibition of tumor growth and the HER2 aptamers conjugated to the drugs presumably enhance the anti-cancer therapeutic efficacy of DM1.

DM1 is known to be a strong inhibitor for microtubule assembly, yielding inherent systemic toxicity [38]. However, the ApDC and free DM1 appeared to be safe at the treated dose, 60 μg DM1/kg. The mice administered with the suggested dose of ApDC or free DM1 did not show any significant body weight loss during treatment (Figure 4B). Aberrant side-effects of DM1 under in vivo conditions were already reported and based on the analyses, preclinical and clinical trials of DM1 conjugates have been allowed and then proceeded [39,40]. Since any systemic cytotoxicity of RNA aptamers has not been reported and only some allergic reaction of PEG molecules has been reported [41], the enhanced anti-cancer therapeutic efficacy would certainly result from the inherent toxicity of DM1 which were more efficiently delivered to the cancer cells. However, at this moment it is uncertain whether providing HER2 targetability and PEG conjugation to DM1 may reduce the systemic toxicity of the drug. Upscaled animal studies would further verify the enhanced safety of the ApDC in vivo.

### 2.5. Histological Changes by ApDC Treatment

At the end point of tumor growth analysis, major internal organs such as tumor, heart, liver, lungs, and spleen were resected, and any forms of cell death were then analyzed through hematoxylin and eosin (H&E) staining and terminal deoxynucleotidyl transferase dUTP nick end labeling (TUNEL) assay. The TUNEL assay verified some apoptotic cells in the superficial area of the tumor tissue of the ApDC-administered mice (Figure 5). In the case of mice treated with free DM1, apoptotic cells were found in tumor tissues and spleen as well.

According to the microscopic analysis, no significant infiltration of immune cells was found in the tissues of major organs of mice treated with the ApDC or free DM1 at 60 μg DM1/kg (Appendix A). These results imply that the intravenous administration of the target specific ApDC appeared to be rather a safer procedure to treat cancer, compared to the direct administration of free DM1. The tumor-targeting aptamer conjugated to DM1 molecule presumably enhanced DM1 accumulation in the tumor tissue. Meanwhile, the free DM1 happened to be more off-targeted to spleen resulting in aberrant tissue damages, due to a lack of tumor-targeting capability.

### 2.6. Changes of Hematological and Biochemical Parameters by ApDC Treatment

In order to further verify the in vivo toxicity of HER2 aptamer-DM1 conjugate, mice were intravenously injected with varied concentrations of free DM1 and the ApDC. Two days later, the hematological and biochemical parameters were analyzed using the blood infraorbital taken. According to the hematological analysis, the counts of platelets, RBC, and hematocrit were not significantly changed at the test dose, 60 μg DM1/kg and the equivalent dose of ApDC (2.7 mg/kg). Any significant changes in the numbers of white blood cells including monocytes, eosinophils, and basophils were not monitored either. These imply that the administration of ApDC and free DM1 at the controlled dose does not elicit serious side-effects such as aberrant immune stimulation and anemia, reported major side-effects of DM1 [39,40]. The biochemical analysis (Table 1) also showed that the administration of ApDC and free DM1 at the dose of 60 μg DM1/kg did not seriously affect the hepatic functions and hematologic characteristics of a mouse. These results suggest that the ApDC and DM1 can be systemically utilized for cancer treatment at the controlled doses.

A reducing environment of the body may readily break the disulfide bonding between the aptamer and free DM1 [30,31,32] and the premature exposure of free DM1 in the blood circulation may cause the aberrant side-effects of DM1. Therefore, when combining a delivery carrier with a drug, it is important to consider the stability of the chemical linkage in the body [42,43,44,45]. The disulfide linkage for the combination of aptamer molecules and drugs can enhance the efficiency of the drug but may not guarantee the stability of the conjugates in the body. In order to make a further analysis regarding the in vivo side-effects of ApDC, the distribution and pharmacokinetics of the drug in the body should be analyzed. A more systematic analysis on a larger scale would tell us a genuine benefit of ApDC in terms of efficacy and toxicity, differentiating from those of free DM1. At the same time, to expand the utilization of aptamer in drug targeting, it is necessary to prove beneficial merits of aptamer-mediated targeting over antibody-mediated drug delivery.

## 3. Materials and Methods

### 3.1. Cell Lines and Cell Culture

BT-474 (breast mammary gland, human), MDA-MB-231 (breast mammary gland, human), MCF-7 (breast adenocarcinoma, human), and A549 (lung carcinoma, human) were purchased from the Korean Cell Line Bank (Seoul, Korea). Following the suppliers’ instructions, the cells were cultured in RPMI media with l-glutamine, 25 mM HEPES supplemented with 10% fetal bovine serum (FBS), and penicillin-streptomycin sulfate at 37 °C in a 5% CO_2_ incubator.

### 3.2. Aptamer and Chemicals

The DNA template of HER2 RNA aptamer and FITC-labeled RNA aptamer were purchased from Bioneer (Daejeon, Korea). Double-stranded DNA templates for transcription were generated via PCR, and the dsDNA templates were purified with a QIAquick PCR purification kit (Qiagen, Valencia, CA, USA). HER2 RNA aptamers were in vitro transcribed from the purified PCR product by a DuraScribe T7 transcription kit (Epicentre, Madison, WI, USA). RNase A- resistant RNA was accomplished by replacing CTP and UTP with 2′-fluorine-dCTP (2′-F-dCTP) and 2′-fluorine-dUTP (2′-F-dUTP) in the DuraScribe in vitro transcription reaction [17]. The sequence of RNA aptamer was 5′-AGC CGC GAG GGG AGG GAU AGG GUA GGG CGC GGC U-3′ with 2′-fluorinated pyrimidines. Mertansine, also called DM1 compound, was purchased from Cayman Chemical (Ann Arbor, MI, USA).

For the determination of aptamer dissociation constant, the HER2 aptamers were diluted to several different concentrations (from 0 to 100 nM) and incubated in the presence of His-tagged HER2 protein (3 nM) in the 200 μL of binding buffer (30 mM Tris-HCl, 150 mM NaCl, 1.5 mM MgCl_2_) for 30 min at room temperature. Ni-NTA agarose beads (Qiagen) were blocked with a buffer (30 mM Tris-HCl, 150 mM NaCl, 1.5 mM MgCl_2_, 1% BSA) for 1 h, washed with a binding buffer (30 mM Tris-HCl, 150 mM NaCl, 1.5 mM MgCl_2_) and then treated with the aptamer-protein mixture for 1 h. The aptamers were eluted by heating the beads to 95 °C and then quantified by real-time q-PCR: 50 °C for 3 min, 95 °C for 2 min, and 40 cycles of 95 °C for 15 s, 60 °C for 30 s, and 72 °C for 30 s (Appendix A). The dissociation constants (Kd) of aptamers were obtained from the binding curves created with GraphPad Prism 5.0 software (GraphPad, San Diego, CA, USA).

### 3.3. Preparation of HER2 Aptamer-DM1 Conjugate

The HER2 aptamer-DM1 conjugate is comprised of HER2 aptamer, DM1, and PEG (polyethylene glycol, 20 kDa) with a bifunctional linker. Aptamer and DM1 were conjugated with a disulfide bond using a succinimidyl 3-(2-pyridyldithio)propionate (SPDP) bifunctional linker. The synthesized HER2 aptamer-DM1-PEG conjugates (Bio-Synthesis, Lewisville, TX, USA), referred to as ApDC, were stored at −20 °C until usage.

### 3.4. Serum Stability of HER2 RNA Aptamer

The HER2 RNA aptamers were incubated in RPMI 1640 media including 10% fetal bovine serum at 37 °C for 0 ~ 72 h and analyzed in 2% agarose gel. The shifted RNAs were stained with Loading Star (Dynebio, Seongnam, Korea), quantified using Gel Quant NET software (BiochemLabSolutions.com), and plotted with GraphPad Prism 5.0 software (GraphPad, San Diego, CA, USA).

### 3.5. Confocal Image Analysis

BT474, MDA-MB-231, MCF-7, and A549 cells (each 1.0 × 10^6^) were seeded on a 4-chamber cell culture plate (SPL life sciences, Pocheon, Korea) and incubated in a CO_2_ incubator at 37 °C for 24 h. After washing with a buffer (30 mM Tris-HCl, 150 mM NaCl, and 1.5 mM MgCl_2_), the cells were treated with FITC-labeled RNA aptamers (25 nM) and incubated at 37 °C for 30 min to 4 h with gentle shaking. The treated cells were washed, stained with Antifade Mounting Medium with DAPI (Vector laboratories, Burlingame, CA, USA) for 30 min and then visualized with a confocal microscope (LSM800, Carl Zeiss, Jena, Germany).

### 3.6. Cytotoxicity Assay

To verify the cytotoxicity of ApDC, BT474 (HER2-positive) and MDA-MB-231 (HER2-negative) cells were seeded in 96-well plates and cultured for 24 h. The plated cells were treated with HER2 aptamer, DM1, or ApDC (0 to 500 nM, 100 μL, *n* = 8) in serum-free culture medium at 37 °C for 4 h. Fresh 20% FBS-containing media (100 μL) were added to the media which were further incubated for 72 h. For live cell counting, 10 μL of WST solution (EZ-cytox, Seoul, Korea) was added to each well which was additionally incubated for 4 h. The absorbance of media in the wells was measured at 450 nm wavelength using Tecan Sunrise microplate reader (Tecan, Männedorf, Switzerland).

### 3.7. In Vivo Xenograft Mouse Model

The animal experiments were performed with the approval of the Institutional Animal Care and Ethics Committee of Yonsei University at Wonju College of Medicine (approval number: YWCI-201808-012-02). BT-474 cells were suspended at a concentration of 1.5 × 10^7^ cells in 200 μL of serum-free media containing 50% of Matrigel (BD Biosciences, Bedford, MA, USA) and subcutaneously injected to the right flank of 7-week-old female BALB/c nude mice (Orient Bio, Seongnam, Korea).

### 3.8. Analyses of In Vivo Anticancer Activity of ApDC

The anticancer therapeutic activity of the HER2 aptamer-DM1-PEG conjugate was evaluated in the mice bearing BT-474 xenografts. When the volumes of implanted tumors reached approximately 100 mm^3^, BT-474 tumor-bearing mice were randomly separated into five groups (*n* = 4, group I: PBS-treated, group II: free DM1-treated, group III: ApDC-treated). DM1 and ApDC were dissolved in PBS and then intravenously administered into mice (60 μg of DM1/kg each) three times every three days. Body weights and tumor volumes were measured every other day (tumor volume = a × b^2^ × 0.52, a; long diameter, b; short diameter). The mice were sacrificed on day 36, and the major organs including tumors were excised, fixed, and embedded in paraffin for histological analysis.

### 3.9. Histological Analysis

On day 35 post injection, the tumor, liver, lungs, spleen, and heart of all treated groups (PBS: PBS-treated, DM1 low: 12 μg of DM1/kg each-treated, DM1 medium: 60 μg of DM1/kg each-treated, DM1 high: 300 μg of DM1/kg each-treated, ApDC: 60 μg of DM1/kg each-treated) were dissected, fixed in 4% formalin, embedded in paraffin, and sectioned at 5 μm thickness. The sections were deparaffinized, hydrated, and stained with H&E reagent for histological analysis. Apoptotic cells in tumor tissues were stained with Click-it^TM^ TUNEL colorimetric kit (Invitrogen, Carlsbad, CA, USA) according to the manufacturer’s instruction. TUNEL-positive cells were manually quantified.

### 3.10. Analysis of Dematological and Biochemical Parameters

To analyze the in vivo toxicity of HER2 ApDC, the hematological and biochemical parameters of the BALB/c normal mice were measured after intravenous administration of a single dose of PBS, free DM1 (60 μg/kg), ApDC (2.7 mg/kg, equivalent to 60 μg of DM1/kg). Subsequently, blood was collected in an EDTA-coated microtainer tube (BD Biosciences, Franklin Lakes, NJ, USA) for complete blood counts (CBC) from the intraorbital vein under isoflurane anesthesia on day 3 and then centrifuged at 1800× *g* for 10 min. Hematological parameters such as counts of platelets, reticulocytes, lymphocytes, and neutrophils were measured with HEMAVET 950 hematology system (Drew Scientific, Waterbury, CT, USA). Mouse sera were also obtained after centrifugation of the blood at 2000× *g* for 20 min. The biochemical parameters of the treated mouse such as serum alanine aminotransferase (ALT), aspartate aminotransferase (AST), alkaline phosphatase (ALP), gamma-glutamyl transpeptidase, and total bilirubin sera were measured with an automated biochemistry analyzer (Konelab 20XT, Thermo Fisher Scientific, Waltham, MA, USA).

### 3.11. Statistical Analysis

The statistical significance of experimental results was determined by the Student’s t test and ANOVA using GraphPad Prism software (GraphPad, San Diego, CA, USA).

## 4. Conclusions

This study shows a representative application of aptamer-based targeted delivery of therapeutics for the treatment of HER2-overexpressing breast cancer. According to the results of the cell-specific binding, cytotoxicity, and in vivo evaluation of the aptamer-drug conjugate ApDC, the HER2 aptamer conjugated to anticancer drug DM1 was able to specifically recognize the target cancer cells and therefore enhance the therapeutic efficacy of the drug. This report provides strong evidence supporting a platform technology of aptamer-guided therapeutics and drug delivery.

## Figures and Tables

**Figure 1 ijms-21-09764-f001:**
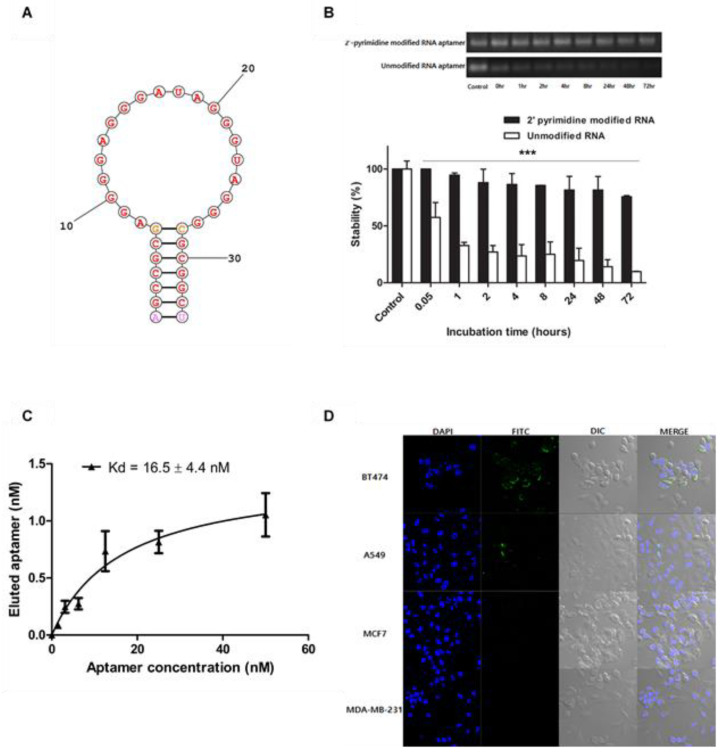
The structure and characterization of HER2 aptamer. (**A**) The secondary structure of HER2 RNA aptamer was estimated using the secondary structure prediction service of Mathews Lab. (**B**) Unmodified RNA aptamer and 2′-pyrimidine-modified RNA aptamer incubated in 10% FBS-containing medium were analyzed by gel retardation analysis (*** *p* < 0.001). (**C**) The dissociation constant of HER2 RNA aptamer was performed using the bead binding assay and analyzed by qPCR. (**D**) In vitro cell bindings of FITC-labeled HER2 RNA aptamer to BT-474, A549, MCF7, and MDA-MB-231 cells were analyzed using confocal microscopy.

**Figure 2 ijms-21-09764-f002:**
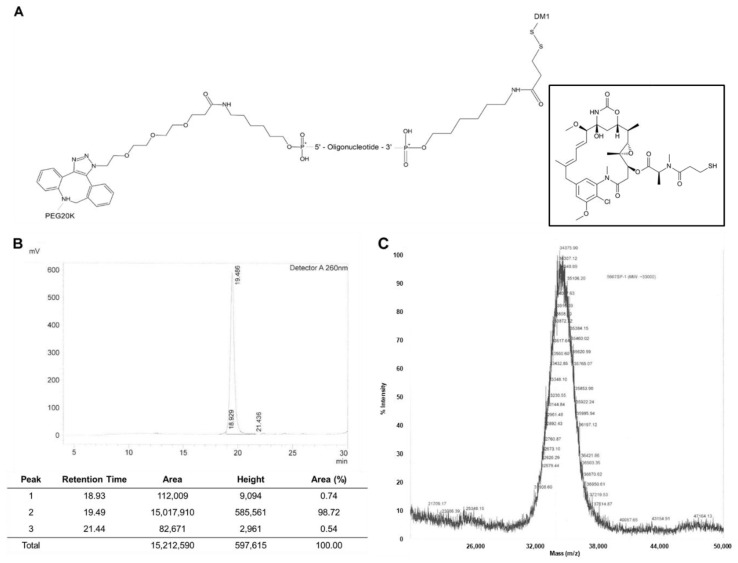
Structure and analysis of HER2 aptamer-DM1 conjugate. (**A**) The aptamer-DM1 conjugate consists of polyethylene glycol (20kDa) and DM1 coupled to the 5′ end and 3′ end of HER2 aptamer, respectively. The HER2 aptamer-DM1 conjugate was purified and analyzed by HPLC (**B**) and mass spectrometry (**C**).

**Figure 3 ijms-21-09764-f003:**
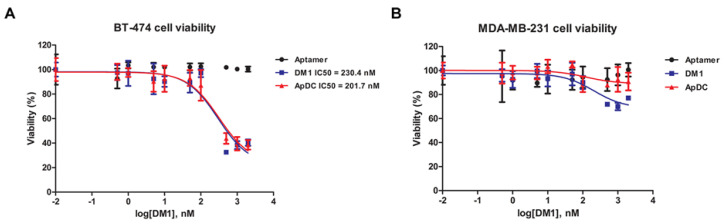
The in vitro cytotoxicity of HER2 aptamer-DM1 conjugate. BT-474 (**A**) or MDA-MB-231 (**B**) cells were treated with the HER2 aptamer-DM1 conjugate and their viability was analyzed using WST-1 assay 72 h later.

**Figure 4 ijms-21-09764-f004:**
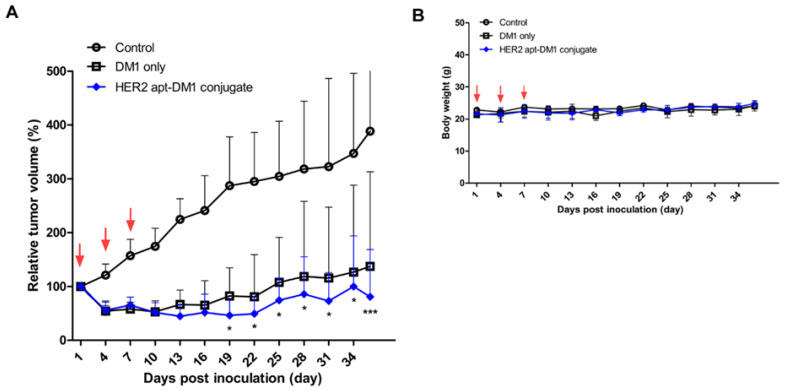
The in vivo tumor growth inhibition by HER2 aptamer-DM1 conjugate. BT-474 cells were subcutaneously injected into the right flank of nude mice. When the size of tumor reached a volume of ~100 mm^3^, the mice began to be treated with the ApDC or free DM1. The mice were treated trice every three days, and the tumor size (**A**) and mouse body weight (**B**) were measured every other day. Red arrows indicate the injection days. * *p* < 0.05, *** *p* < 0.001.

**Figure 5 ijms-21-09764-f005:**
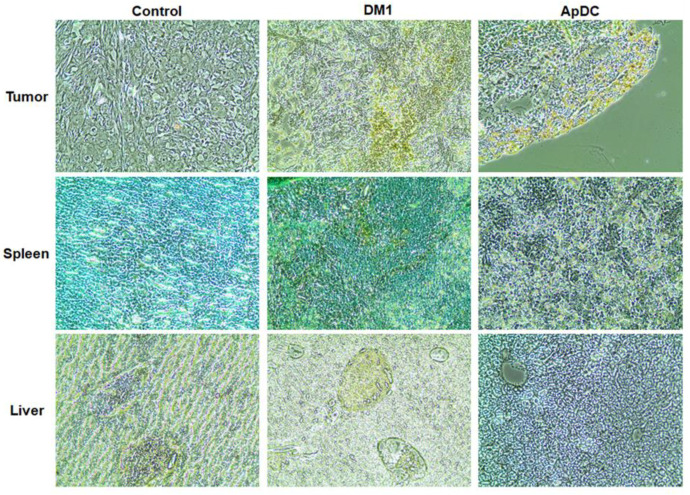
Histochemical changes by the administration of HER2 aptamer-DM1 conjugate. The tissues of tumor, spleen, and liver were stained with a terminal deoxynucleotidyl transferase dUTP nick end labeling (TUNEL) kit on the final day of in vivo study and examined with a microscope (200×).

**Table 1 ijms-21-09764-t001:** Hematological and biochemical parameters measured after ApDC treatment (*n* = 3).

		Control (PBS)		DM1 (60 μg/kg)		ApDC (3.85 mg/kg)	
		Mean	SD	Mean	SD	Mean	SD
Hematology, 10^3^/μL	Platelet, counts	697.50	64.16	632.40	101.64	425.33	66.11
	Red blood cells, M/μL	8.38	0.27	8.08	0.35	7.69	0.34
	Hematocrits, %	45.86	2.37	46.38	1.41	43.17	1.20
	Neutrophil, absolute	0.54	0.09	0.37	0.15	0.31	0.11
	Lymphocyte	2.22	0.50	1.78	0.89	1.55	0.18
Clinical chemistry, U/L	ALT	26.28	6.33	32.29	14.33	42.93	14.86
	AST	89.46	17.86	89.20	11.66	95.35	20.89
	ALP	295.11	99.65	292.01	84.44	211.76	44.13
	GGT	4.97	1.00	7.02	2.11	6.28	3.67
	Total bilirubin	0.06	0.03	0.07	0.03	0.07	0.02

Hematological parameters and biochemical parameters were measured with a hematology system and an automated biochemistry analyzer 3 days after inoculation, respectively.

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
