# Peer review of "Development of HER2-Specific Aptamer-Drug Conjugate for Breast Cancer Therapy"

_ijms, 2020, doi:10.3390/ijms21249764_

Round 1

Reviewer 1 Report

An interesting study where the anti-cancer effectiveness of a HER2 RNA aptamers - mertansine conjugate is evaluated.  While they show some promising results, there are a few major flaws that must be addressed before publication.

1) The authors describe the disulfide linkage that holds the DMI to the aptamer as pH-cleavable.  They say the disulfide bond would be easily cleaved in the low pH environment of endocytic vesicles and reference a Levy paper (30) that has nothing to do with this comment. They also describe that the linkage can be broken in a "reducible" environment of endosomes.  Neither of these are correct.  Disulfides can be reduced and cleaved in high pH environments, not low pH environments.  Also, reducible and reducing are opposite terms - if something is reducible then it is oxidizing.  This needs to be addressed.

2) The hypothesis of the work is that the linkage fragility of the conjugate would facilitate the release of the drug from the endosome which would elevating the anticancer therapeutic efficacy of the drug.  There is no control of a non-cleavable linker holding the aptamer and the drug to support this idea.  If this aspect of the study is removed then it is not very original in comparison to the work of Weihong Tan where Maytansine is tethered to the HER2 aptamer (https://pubs.acs.org/doi/full/10.1021/acs.bioconjchem.0c00250) 

3) Importantly, it should be noted that PEGylation of the aptamer can lead to allergic reactions (see https://www.ncbi.nlm.nih.gov/pmc/articles/PMC5819876/) and this should be mentioned

Author Response

Dear Reviewer 1

Enclosed, please find a revised version of our manuscript entitled, “Development of HER2-specific aptamer-drug conjugate for breast cancer therapy” (Manuscript ID: ijms-1016963). The manuscript has been revised in accord with the Reviewers’ suggestions as summarized below. Changes are marked in red here and in the revised manuscript.

Overall comment.

An interesting study where the anti-cancer effectiveness of a HER2 RNA aptamers - mertansine conjugate is evaluated.  While they show some promising results, there are a few major flaws that must be addressed before publication.

Comment 1.

The authors describe the disulfide linkage that holds the DMI to the aptamer as pH-cleavable.  They say the disulfide bond would be easily cleaved in the low pH environment of endocytic vesicles and reference a Levy paper (30) that has nothing to do with this comment. They also describe that the linkage can be broken in a "reducible" environment of endosomes.  Neither of these are correct.  Disulfides can be reduced and cleaved in high pH environments, not low pH environments.  Also, reducible and reducing are opposite terms - if something is reducible then it is oxidizing. This needs to be addressed.

Response:

Thanks for your suggestion. Reviewer 1 pointed out a serious mistake in referring to an article regarding pH-dependent cleavage of a disulfide bond. The misplaced reference (30) was moved to #33 and two more reports (#31,32) were additionally included to provide more information about conjugation via a disulfide linkage. As suggested, the term of reducible was changed to reducing.

Revised manuscript:

“In this study, we developed a HER2-specific aptamer-drug conjugate for targeted cancer therapy. The HER2-specific aptamer was conjugated with anti-cancer mertansine (DM1) by a pH-dependent cleavable disulfide bond at the 3' end of aptamer [30-32]. A polyethylene glycol molecule (M.W. 20 kDa) was also conjugated to the 5' end of aptamer for prolonged circulation in vivo. The HER2-specific ApDC was tested in HER2-positive cancer cells to verify its target-specific binding capability and cytotoxicity. Also, the ApDC was systemically administered into mice carrying tumor xenografts to evaluate its targeted anti-cancer therapeutic efficacy.”

#31) Saito, G.; Swanson, J. A.; Lee, K. D. Drug delivery strategy utilizing conjugation via reversible disulfide linkages: role and site of cellular reducing activities. Adv Drug Deliv Rev. 2003;55(2):199-215.

#32) Bauhuber, S.; Hozsa, C.; Breunig, M.; Gopferich, A. Delivery of nucleic acids via disulfide-based carrier systems. Adv Mater. 2009;21(32-33):3286-3306.

“However, the stability of RNA can be secured by various conventional modification methods, and the RNA aptamer used in the present study (Figure 1A) was also modified at the pyrimidine by 2’-fluorinated pyrimidine, thereby achieving much higher stability than the unmodified RNA aptamer [33].”

Comment 2.

The hypothesis of the work is that the linkage fragility of the conjugate would facilitate the release of the drug from the endosome which would elevating the anticancer therapeutic efficacy of the drug.  There is no control of a non-cleavable linker holding the aptamer and the drug to support this idea.  If this aspect of the study is removed then it is not very original in comparison to the work of Weihong Tan where Maytansine is tethered to the HER2 aptamer (https://pubs.acs.org/doi/full/10.1021/acs.bioconjchem.0c00250)

Response:

The reviewer suggested that evaluation of the anticancer therapeutic efficacy with a compound of non-cleavable linker would further support the significance of this work. Since we do not have the data with non-cleavable control compounds at this moment, two more references (#34, 35) reporting cleavage of disulfide linkage in the cell were included.

Revised manuscript:

“The linkage fragility of ApDC would facilitate the release of DM1 from the endocytic pathway, elevating the anticancer therapeutic efficacy of the drug [30, 34, 35].

“The disulfide bond would be easily cleaved in the low pH environment of endocytic vesicles, resulting in facilitated escapes of the freed DM1 from endosomes [30, 34, 35].”

#34) Yang, Q.; Deng, Z.; Wang, D.; He, J.; Zhang, D.; Tan, Y.; Peng, T.; Wang, X. Q.; Tan, W. Conjugating Aptamer and Mitomycin C with Reductant-Responsive Linker Leading to Synergistically Enhanced Anticancer Effect. J Am Chem Soc. 2020;142(5):2532-2540.

#35) Erickson, H. K.; Widdison, W. C.; Mayo, M. F.; Whiteman, K.; Audette, C.; Wilhelm, S. D.; Singh, R. Tumor delivery and in vivo processing of disulfide-linked and thioether-linked antibody-maytansinoid conjugates. Bioconjug Chem. 2010;21(1):84-92.

Comment 3.

Importantly, it should be noted that PEGylation of the aptamer can lead to allergic reactions (see https://www.ncbi.nlm.nih.gov/pmc/articles/PMC5819876/) and this should be mentioned.

Response:

As suggested, concern about possible allergic reactions of PEG molecules was stated (L180P6).

Revised manuscript:

“Since any systemic cytotoxicity of RNA aptamers has not been reported and only some allergic reaction of PEG molecules has been reported [41], the enhanced anti-cancer therapeutic efficacy would certainly result from the inherent toxicity of DM1 which were more efficiently delivered to the cancer cells”

#41) Ganson, N. J.; Povsic, T. J.; Sullenger, B. A.; Alexander, J. H.; Zelenkofske, S. L.; Sailstad, J. M.; Rusconi, C. P.; Hershfield, M. S. Pre-existing anti-polyethylene glycol antibody linked to first-exposure allergic reactions to pegnivacogin, a PEGylated RNA aptamer. J Allergy Clin Immunol. 2016;137(5):1610-1613e7.

We hope that we have satisfactorily responded to all of the questions/comments made by the Reviewers, and that the manuscript is now suitable for publication in International Journal of Molecular Sciences. We are quite excited by the prospect, so please do not hesitate to contact me if you have additional questions or problems. As always, we look forward to publishing additional manuscripts with you, and remain,

Sincerely,

Yong Serk Park, Ph.D., Professor

Dept. Biomedical Laboratory Science

Yonsei University

Wonju, Gangwon 220-710

Republic of Korea

TEL: +82-371-760-2448

FAX: +82-371-763-5224

Email: parkys@yonsei.ac.kr

*Please see the revised manuscript in the attachment

Reviewer 2 Report

It is  a valuable piece of information which could be improved by more detailed in vivo research.

All questions to your study are indicated as comments to the highlighted text.

One controversy for me in this paper, or better too say incomprehensibleness is the use of the human lung cancer cell line A549. Why did you choose this line instead, for example, SKBR-3 breast cancer cell line, HER2+  ? It would be better to provide results for SKBR-3 or simply remove the results for A549, as it is not a breast cancer model and I couldn’t find any explanation confirming the use of A549.

Good luck in further studies improving the results described here!

Author Response

Dear Reviewer 2

Enclosed, please find a revised version of our manuscript entitled, “Development of HER2-specific aptamer-drug conjugate for breast cancer therapy” (Manuscript ID: ijms-1016963). The manuscript has been revised in accord with the Reviewers’ suggestions as summarized below. Changes are marked in red here and in the revised manuscript.

Overall comment.

It is a valuable piece of information which could be improved by more detailed in vivo research.

Comment 1.

All questions to your study are indicated as comments to the highlighted text.

Response:

Thanks for your comments. As suggested, we corrected the manuscript.

Revised manuscript:

Compared to the control group, the mice treated with the ApDC showed a significant reduction of tumor growth. Besides, any significant body weight losses or hepatic toxicities were monitored in the ApDC-treated mice.”

“Separately, various structures of aptamers have been developed as targeting ligands.”

“The target-specific cytotoxicity of HER2 aptamer-DM1 conjugate was analyzed by the WST-1 assay (n=8) on HER2-positive BT-474 and HER2-negative MDA-MB-231 cancer cells.”

“Two days later hematological and biochemical parameters were analyzed using the blood infraorbital taken.”

“Table 1. Hematological and biochemical parameters measured after ApDC treatment (n=3)*”

“To verify the cytotoxicity of ApDC, BT474 (HER2-positive) and MDA-MB-231 (HER2-negative) cells were seeded in 96-well plates and cultured for 24 h. The plated cells were treated with HER2 aptamer, DM1, or ApDC (0 to 500 nM, 100 μL, n=8) in serum-free culture medium at 37°C for 4 h.”

“On day 35 post injection, the tumor, liver, lungs, spleen, and heart of all treated groups (PBS: PBS-treated, DM1 low: 12 μg of DM1/kg each-treated, DM1 medium: 60 μg of DM1/kg each-treated, DM1 high: 300 μg of DM1/kg each-treated, ApDC: 60 μg of DM1/kg each-treated) were dissected, fixed in 4% formalin, embedded in paraffin, and sectioned at 5 μm thickness.”

Comment 2.

One controversy for me in this paper, or better too say incomprehensibleness is the use of the human lung cancer cell line A549. Why did you choose this line instead, for example, SKBR-3 breast cancer cell line, HER2+ ? It would be better to provide results for SKBR-3 or simply remove the results for A549, as it is not a breast cancer model and I couldn’t find any explanation confirming the use of A549.

Response:

Thanks for your suggestion. Reviewer 2 claimed that the selection of human lung cell line A549 as a control cell line for in vitro analysis. Throughout the manuscript, two different breast cancer cell lines (HER2-positive BT-474 and HER2-negative MDA-MB-231 cells) were tested. In order to verify the specific interactions between aptamer molecules and HER2 receptors, another cancer cell line (A549 human lung carcinoma) was included in in-vitro cell-binding analyses.

We hope that we have satisfactorily responded to all of the questions/comments made by the Reviewers, and that the manuscript is now suitable for publication in International Journal of Molecular Sciences. We are quite excited by the prospect, so please do not hesitate to contact me if you have additional questions or problems. As always, we look forward to publishing additional manuscripts with you, and remain,

Sincerely,

Yong Serk Park, Ph.D., Professor

Dept. Biomedical Laboratory Science

Yonsei University

Wonju, Gangwon 220-710

Republic of Korea

TEL: +82-371-760-2448

FAX: +82-371-763-5224

Email: parkys@yonsei.ac.kr

*Please see the revised manuscript in the attachment.

Reviewer 3 Report

Article is very interesting in the field of aptamer-drug conjugates for (anticancer) drug delivery. However, manuscript can be further improved following the above minor comments:

1) In Figure 2, the complete structure of DM1 should be drawn.

2) A more detailed and clear explanation about the origin of HER2 aptamer should be added to the manuscript. Has HER2 aptamer been selected by the same authors by SELEX? If so, a description of SELEX procedure has to be added to Results and Discussion. On the other hand, if HER2 aptamer was already present in the literature, related information have to be added to Introduction.

3) Authors stated the importance of deep knowledge on aptamer structures. Circular dichroism spectra of HER2 aptamer alone as well as its conjugate with DM1 should be added to section 2.1. and 2.2., respectively, and discussed. The structure reported in Figure 1A could not be a good representation of the real structure of this aptamer and recording CD spectra could be an easy way to obtain further information on the actual secondary structure of the aptamer (duplex or G-quadruplex?). Moreover, a comparison of CD spectra of HER2 aptamer alone and its conjugate would add precious information on putative conformational changes of the secondary structure of the aptamer in the absence and presence of the drug.

Author Response

Dear Reviewer 3

Enclosed, please find a revised version of our manuscript entitled, “Development of HER2-specific aptamer-drug conjugate for breast cancer therapy” (Manuscript ID: ijms-1016963). The manuscript has been revised in accord with the Reviewers’ suggestions as summarized below. Changes are marked in red here and in the revised manuscript.

Overall comment.

Article is very interesting in the field of aptamer-drug conjugates for (anticancer) drug delivery. However, manuscript can be further improved following the above minor comments.

Comment 1.

In Figure 2, the complete structure of DM1 should be drawn.

Response:

Thanks for your helpful suggestions. As suggested, the structure of DM1 was included (Figure 2).

Revised figure:

Comment 2.

A more detailed and clear explanation about the origin of HER2 aptamer should be added to the manuscript. Has HER2 aptamer been selected by the same authors by SELEX? If so, a description of SELEX procedure has to be added to Results and Discussion. On the other hand, if HER2 aptamer was already present in the literature, related information have to be added to Introduction.

Response:

A reference to the information about the HER2 aptamer was noted (reference #29).

Revised manuscript:

“Several anti-ErbB2 aptamers were already developed as an alternative therapeutic modality [28]. The anti-ErbB2 RNA aptamer was used in this study for ApDC development [29].”

Comment 3.

Authors stated the importance of deep knowledge on aptamer structures. Circular dichroism spectra of HER2 aptamer alone as well as its conjugate with DM1 should be added to section 2.1. and 2.2., respectively, and discussed. The structure reported in Figure 1A could not be a good representation of the real structure of this aptamer and recording CD spectra could be an easy way to obtain further information on the actual secondary structure of the aptamer (duplex or G-quadruplex?). Moreover, a comparison of CD spectra of HER2 aptamer alone and its conjugate would add precious information on putative conformational changes of the secondary structure of the aptamer in the absence and presence of the drug.

Response:

Thanks for your critiques. As Reviewer 3 mentioned, CD spectra of HER2 aptamer and its DM1 conjugate would give us information about the secondary structures of the compounds. Frankly, we do not have the CD spectra for the compounds at this moment. The HER2-specific targeting of the aptamer-DM1 conjugate (Figure 3) may imply that there are no serious conformation changes in the HER2 aptamer structure after conjugation to DM1.

We hope that we have satisfactorily responded to all of the questions/comments made by the Reviewers, and that the manuscript is now suitable for publication in International Journal of Molecular Sciences. We are quite excited by the prospect, so please do not hesitate to contact me if you have additional questions or problems. As always, we look forward to publishing additional manuscripts with you, and remain,

Sincerely,

Yong Serk Park, Ph.D., Professor

Dept. Biomedical Laboratory Science

Yonsei University

Wonju, Gangwon 220-710

Republic of Korea

TEL: +82-371-760-2448

FAX: +82-371-763-5224

Email: parkys@yonsei.ac.kr

*Please see the revised manuscript in the attachment.

Round 2

Reviewer 1 Report

The manuscript has been improved.  However the authors still state that the disulfide bond will be cleaved in the low pH environment - this is incorrect and must be changed.  Also the paper by Weihong Tan should be included and the originality of this paper with respect to that one should be stressed.

Author Response

Dear Sir

Thank you very much for your kind revision.

I attached the response for your comment.

Thank you again and have a nice day.

Kind regards,
